# A global assessment of the drivers of threatened terrestrial species richness

Christine Howard [1✉], Curtis H. Flather [2] & Philip A. Stephens [1]

High numbers of threatened species might be expected to occur where overall species richness is also high; however, this explains only a proportion of the global variation in threatened species richness. Understanding why many areas have more or fewer threatened species than would be expected given their species richness, and whether that is consistent across taxa, is essential for identifying global conservation priorities. Here, we show that, after controlling for species richness, environmental factors, such as temperature and insularity, are typically more important than human impacts for explaining spatial variation in global threatened species richness. Human impacts, nevertheless, have an important role, with relationships varying between vertebrate groups and zoogeographic regions. Understanding this variation provides a framework for establishing global conservation priorities, identifying those regions where species are inherently more vulnerable to the effects of threatening human processes, and forecasting how threatened species might be distributed in a changing world.

[1] Conservation Ecology Group, Department of Biosciences, Durham University, South Road, Durham DH1 3LE, UK. [2] US Forest Service Rocky Mountain Research Station, Fort Collins, CO 80526, USA. ✉email: christine.howard@durham.ac.uk

Safeguarding species against threatening processes is a major global priority[1–3]. Yet, despite the adoption of numerous international agreements aimed at addressing the biodiversity crisis, biodiversity continues to be lost[4,5]. An ambitious global strategy is now required that clearly defines the actions and policies necessary to restore ecosystems to levels that help both people and nature thrive[6–8]. This will require the establishment of global conservation priorities, so that limited resources can be focussed on those areas of highest conservation value most at risk from environmental change[6,9,10]. Currently, conservation plans often target overall species diversity, under the assumption that it can act as an adequate surrogate for other dimensions of biodiversity[11–15]. Biodiversity indices, however, are seldom spatially congruent[11–13,15–17], and total species richness can only partially explain spatial gradients in threatened species richness (Fig. 1, see methods). Areas of South America, sub-Saharan Africa, India and Southeast Asia all have far greater numbers of threatened species than would be expected given the total size of the species pool. Understanding why these areas are hotspots of imperilment and how the mechanisms driving these patterns differ between taxonomic groups, is key for identifying global conservation priorities and the actions and policies required to prevent further loss of biodiversity[6–8,18]. The processes underlying global patterns of threatened species richness, however, remain largely unexplored.

Extinction is often driven by threatening human activities[19,20]. Increasing human populations, long-term land cover change, and pressures from invasive alien species have all been linked to the loss of biodiversity[21–23]. Environmental conditions can, however, predispose species to the effects of these threatening processes[13]. Energy availability and habitat heterogeneity have been linked with speciation rates via increases in available niche space and retention of rare species through the provision of refugia, and by providing opportunities for isolation and divergent adaptation[24–27]. Threatened species richness is likely driven by the interaction between the predisposing environmental conditions that promote the occurrence of extinction-prone, narrow-range endemic species and the threatening human processes that erode biodiversity[13]. Despite substantial efforts being dedicated to deciphering the determinants of spatial gradients in total species richness[24,26], there is only limited understanding of the drivers of spatial gradients in threatened species richness and how they differ from those of total species richness[28]. If we are to fully understand the distribution of imperilment, we require a global assessment that disentangles the effects of predisposing ecological conditions from those attributable to threatening human processes.

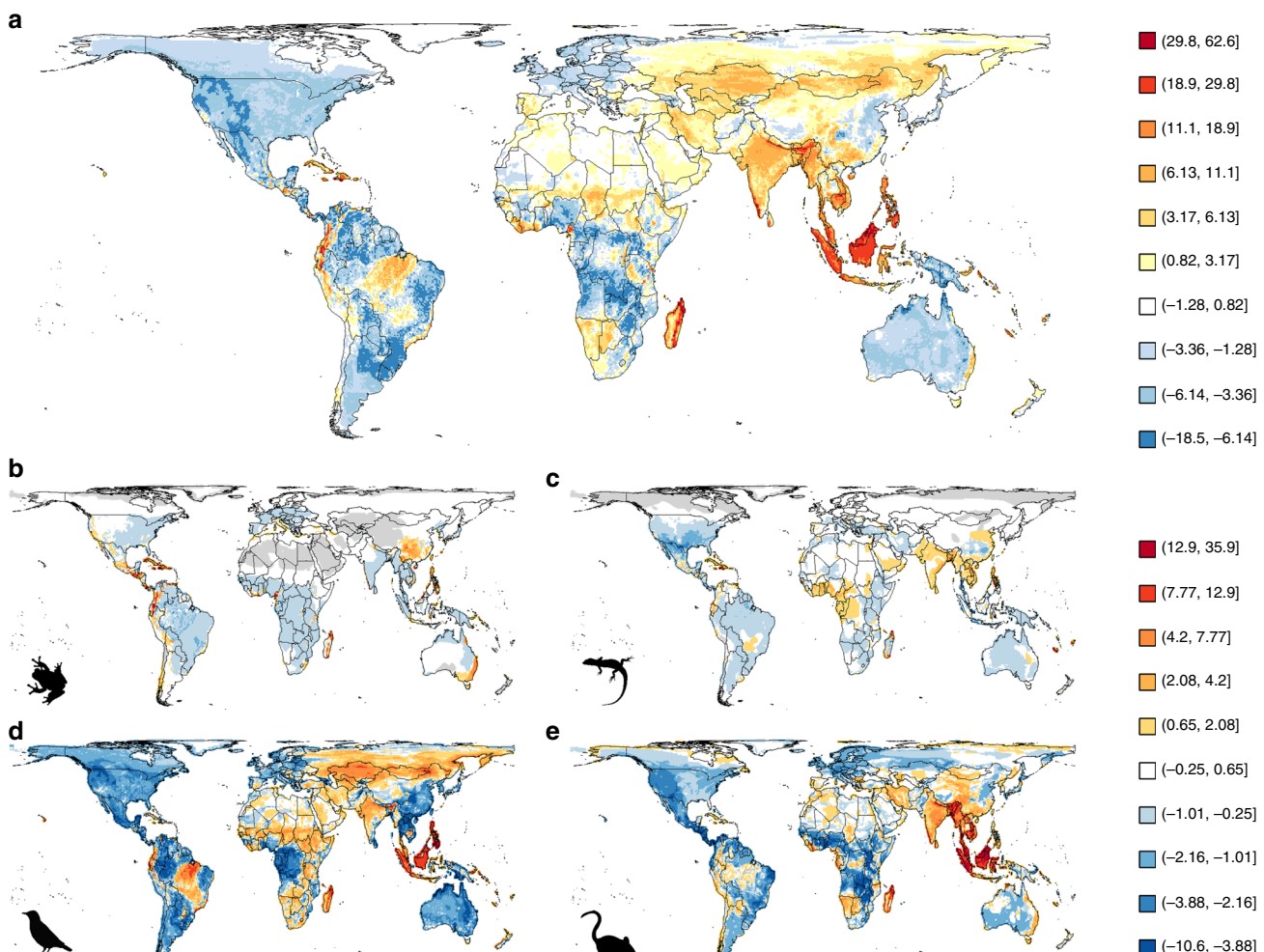

**Fig. 1 Global variation in threatened species richness, having accounted for total species richness.** The maps show the mean residuals from 10 global scale models of threatened species richness when explained by total species richness, for **a** terrestrial vertebrates, and four separate taxonomic groups: **b** amphibians, **c** reptiles, **d** birds and **e** mammals. The colour scale indicates model residuals in terms of the number of threatened species and uses Jenks natural breaks[81] to determine the break points. Note the scales differ between **a**, and **b**–**e**. Grey indicates areas where there are no species classified as threatened. Model performance was moderate (Table 1 and Supplementary Table 2). Source data are provided as a Source Data file.

Here, we present a global assessment of the drivers of threatened species richness patterns, revealing the importance of extrinsic threatening processes and predisposing environmental conditions in shaping concentrations of imperilment. Our models of threatened species richness are based on a data set of 26,746 species of terrestrial vertebrates, 20% of which are considered 'threatened' with extinction (IUCN Red List status of critically endangered, endangered, or vulnerable). Specifically, we quantify the importance of a suite of environmental and human impact covariates, previously shown to correlate with extinction risk and species richness patterns, in driving spatial variation in global threatened vertebrate species richness. To identify the drivers of threatened species richness independent of the drivers of the size of the species pool, we account for the role of total species richness within our models. We then assess the influence of the covariates on four taxonomic groups individually (amphibians, reptiles, birds and mammals) and at the level of individual zoogeographic regions[29] to establish how the drivers of threatened species richness vary between taxa and across space. Finally, we explore how the form of the relationship between these variables and species richness differs when that richness is restricted to threatened species or includes the total species pool.

Our results show that environmental covariates are typically more important than human impacts for explaining global variation in threatened species richness. Nevertheless, we identify substantial variation between zoogeographic regions and taxonomic groups in the importance of individual environmental and human impact variables in driving threatened vertebrate species richness. Specifically, we demonstrate that individual human impact variables can be influential at regional scales. These results suggest that some regions and taxonomic groups may be inherently more vulnerable than others to the effects of threatening human activities, which has important implications for identifying global conservation priorities.

## Results and discussion

**Global drivers of threatened species richness**. Global models of threatened vertebrate species richness revealed a greater influence of environmental parameters than extrinsic human threats (Fig. 2a, Tables 1 and 2), although human impacts still had an important role. Specifically, insularity, or the area of land mass to which a cell belonged, was of the greatest importance for explaining global threatened vertebrate species richness. Island regions generally support a higher number of endemic, extinction-prone species than continental land masses[30] (Supplementary Fig. 10). Two variables associated with the amount of productive energy in a system, namely temperature seasonality and annual precipitation, were also highly influential in our models (ranked 3rd and 4th, respectively, of 16 variables). Given that our models account for total species richness, this result is not a consequence of more productive regions harbouring higher numbers of threatened species simply because they have more species overall. Furthermore, and contrary to previous reports[31], the limited influence of human impact variables at a global scale suggests that this result is unrelated to enhanced human influence in areas of greater net primary productivity[32–34]. Rather, our results suggest that the processes that drive speciation may also be responsible for driving extinction[35]. Areas of high levels of speciation are associated with small range, specialist species that often persist at low population sizes and are therefore vulnerable to demographic stochasticity or extinction from local catastrophes or human disturbance[33,36,37]. Of the human impact variables, those of most importance were long-term land cover change and invasive alien species (ranked 6th and 7th, respectively). These results are in line with previous findings, with habitat loss widely cited as the most

common cause of species' extinctions[38,39], and the impacts of invasive alien species as the second most common cause[40].

**Variation between taxonomic groups**. Models for separate taxa indicated that the dominant processes leading to concentrations of threatened species richness differed substantially between the four vertebrate classes. Specifically, whilst total species richness was the most influential variable for determining the richness of threatened species from all vertebrate classes, the importance of environmental and human impact variables differed substantially between the four groups (Fig. 2). These differences may be driven by varying sensitivities of species to different threatening processes[41]. For example, for amphibians the diversity of elevation was the most influential variable after total species richness (Fig. 2b), with the highest numbers of threatened amphibians occurring in the most topographically diverse areas, see below. Topographic diversity has been linked with increased speciation rates, and the occurrence of small range, endemic species which are more naturally prone to extinction[36,37]. This effect, however, was particularly strong in the South American, Amazonian and Australian regions, reflecting the distribution of Chytridiomycosis, an infectious fungal disease partially responsible for the worldwide declines of amphibian populations and prevalent in the montane habitats of those regions[42]. For birds, temperature seasonality ranked as the most influential variable after total species richness (Fig. 2d), with concentrations of threatened birds occurring in areas of particularly high or low temperature variation, see below. More seasonal environments are often associated with the occurrence of migratory species, which can be more vulnerable than their non-migratory counterparts[43]. Conversely, benign, less seasonal environments may allow the persistence of naturally rare, extinction prone species[44]. Differences in the major processes affecting different taxa may also be linked to dispersal ability. More mobile species tend to be less restricted by local heterogeneities in habitat structure, and more likely to be at equilibrium with climatic conditions[45]. Thus, those habitat variables that vary over a finer resolution, such as topography and human influence, may be more influential in explaining the distribution of threatened amphibians and reptiles, which tend to be less dispersive than birds and mammals.

**Spatial variation in the drivers of threat**. To assess the consistency of drivers of threat across space, we fitted separate models of threatened vertebrate species richness for individual zoogeographic regions[29]. Apart from the universally high importance of total species richness, the importance of individual environmental and human impact variables in driving threatened vertebrate species richness showed substantial variation between zoogeographic regions, as well as from the global model (Fig. 3). In areas where threatened species richness is greater than expected given the total species richness (Fig. 1a), the most influential variables were often related to threatening human processes. For example, long-term land cover change was of the greatest influence in driving threatened vertebrate species richness in the Amazonian region, whilst the area of human-dominated land uses was the most important variable in Madagascar (Fig. 3). This may be driven by the prevalence of small range, endemic species within these regions, which are vulnerable to the effects of habitat loss[33]. In contrast, however, across Southeast Asia, variables related to the environment were of greater importance, with insularity in the Indo-Malayan region, temperature seasonality in the Chinese region and annual precipitation in the Oriental region all emerging as important. The finding that human impact variables can be of greater importance than environmental covariates emerges despite the relatively coarse resolution of our analyses. Climatic

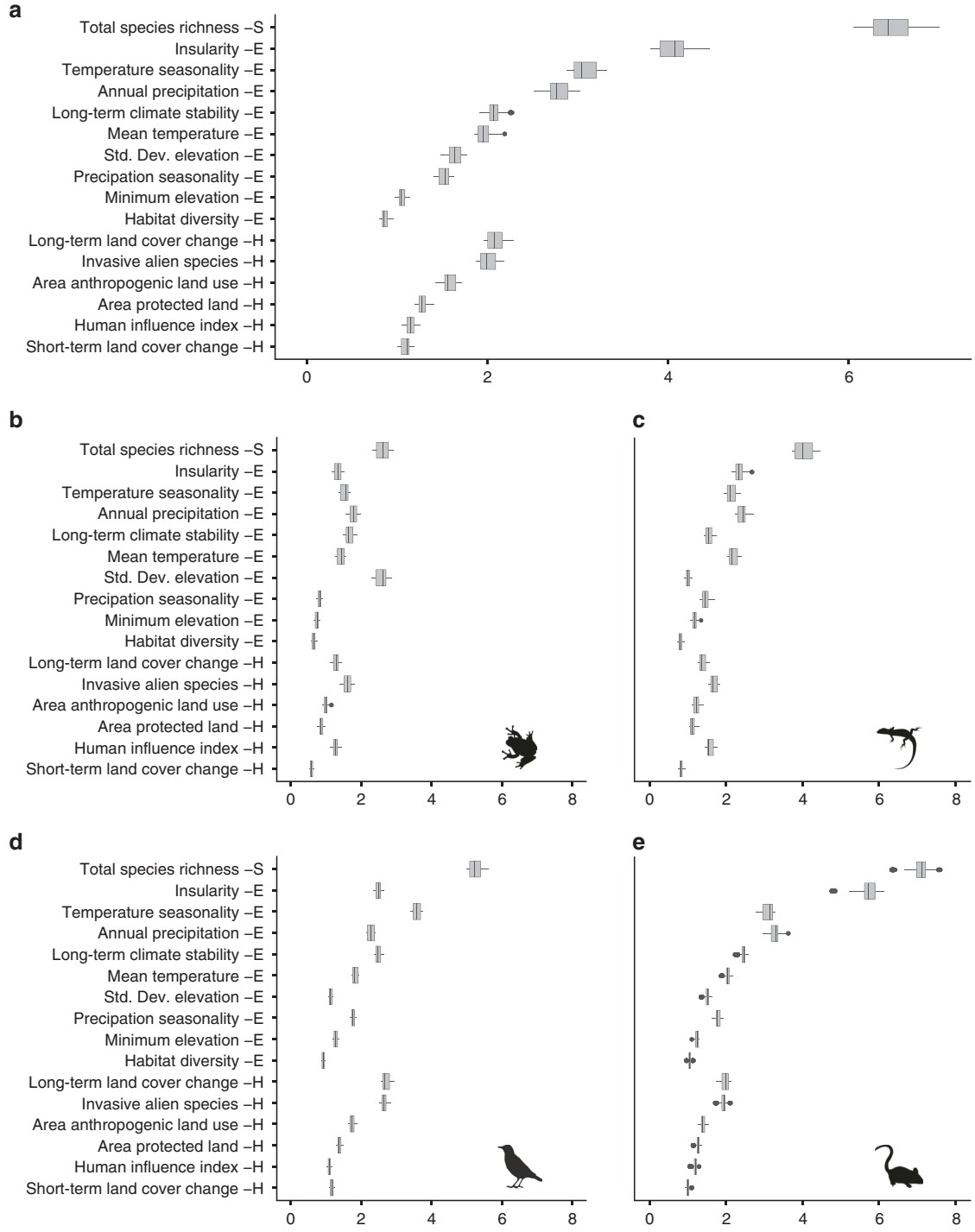

**Fig. 2 Importance of individual variables for predicting global threatened species richness.** Top panel **a** indicates the importance of individual variables from the global models of vertebrate threatened species richness, whilst the bottom panels indicate the measures of individual variable importance from the global models of amphibian (**b**), reptile (**c**), bird (**d**) and mammal (**e**) threatened species richness. Variables are grouped into broader classes, which are indicated by the capital letters on the side of the variable names: Total Species Richness (S), Environmental (E), and Human Impact (H) covariates. Variables are ranked first by groups and then by their median importance in the global model of vertebrate threatened species richness. The vertical line across each box indicates the median and the box boundaries indicate the interquartile range (IQR). Whiskers identify extreme data points that are not more than 1.5 times the IQR on both sides; the dots are more extreme outliers. Source data are provided as a Source Data file.

variables (i.e., temperature and precipitation) can explain a lot of the variation in land cover distribution, including human-related variables[46]. Hence, climatic variables may be identified as being of greater importance than land cover in coarser resolution analyses.

Our large-scale analysis offers important insights into the drivers of broad-scale irreversible changes[47]. Clearly, our global analysis is less well suited to identifying the drivers of fine-resolution changes in threatened species richness.

**Table 1 Performance ($R^2$) of models of threatened species richness.**

|  |  | Total vertebrates | Amphibians | Reptiles | Birds | Mammals |
|---|---|---|---|---|---|---|
| Total species richness only | Global | 0.49 ± 0.03 | 0.04 ± 0.03 | 0.45 ± 0.04 | 0.37 ± 0.08 | 0.36 ± 0.05 |
| Total species richness, environmental and anthropogenic covariates | Global | 0.94 ± 0.01 | 0.72 ± 0.05 | 0.87 ± 0.02 | 0.92 ± 0.01 | 0.94 ± 0.01 |
|  | Regional | 0.81 ± 0.14 | 0.63 ± 0.20 | 0.70 ± 0.19 | 0.77 ± 0.16 | 0.79 ± 0.15 |

Models are fitted at the global scale with only total species richness as a predictor, and at the global and regional scales using total species richness and a range of environmental and human impact covariates. Each datum represents the mean model performance, measured using $R^2$, and standard deviation, across random forest models (mean ± SD). A full summary of model parameters and of regional level model performance can be found in Supplementary Tables 1 and 2. Source data are provided as a Source Data File.

**Table 2 Comparing the importance of environmental and human impact covariates.**

|  |  | Estimate | Standard error | z-value | p-value |
|---|---|---|---|---|---|
| Total vertebrates | Environmental—human impact | 0.58 | 0.05 | 11.64 | <0.01 |
| (ANOVA: $F_{2,18} = 5817.1$, $p < 0.01$) | Total species richness—human impact | 4.93 | 0.05 | 98.69 | <0.01 |
|  | Total species richness—environmental | 4.35 | 0.05 | 87.05 | <0.01 |
| Amphibians | Environmental—human impact | 0.29 | 0.03 | 10.45 | <0.01 |
| (ANOVA: $F_{2,18} = 1673$, $p < 0.01$) | Total species richness —human impact | 1.52 | 0.03 | 54.50 | <0.01 |
|  | Total species richness—environmental | 1.23 | 0.03 | 44.04 | <0.01 |
| Reptiles | Environmental—human impact | 0.38 | 0.04 | 9.42 | <0.01 |
| (ANOVA: $F_{2,18} = 2756.4$, $p < 0.01$) | Total species richness—human impact | 2.73 | 0.04 | 68.49 | <0.01 |
|  | Total species richness—environmental | 2.36 | 0.04 | 59.07 | <0.01 |
| Birds | Environmental—human impact | 0.18 | 0.03 | 5.67 | <0.01 |
| (ANOVA: $F_{2,18} = 7346.1$, $p < 0.01$) | Total species richness—human impact | 3.46 | 0.03 | 107.69 | <0.01 |
|  | Total species richness—environmental | 3.28 | 0.03 | 102.03 | <0.01 |
| Mammals | Environmental—human impact | 0.99 | 0.06 | 15.96 | <0.01 |
| (ANOVA: $F_{2,18} = 4602.9$, $p < 0.01$) | Total species richness—human impact | 5.60 | 0.06 | 89.92 | <0.01 |
|  | Total species richness—environmental | 4.61 | 0.06 | 73.96 | <0.01 |

Repeated measures ANOVAs, and post-hoc analyses, comparing the mean measures of relative importance of environmental and human impact covariates and total species richness in explaining global threatened vertebrate species richness. Source data are provided as a Source Data File.

The lack of congruence in the importance of variables across space could be linked to differences between regions in how much ecological factors vary[23]. For example, in contrast to the global models, insularity was of very low importance in most of the region-level models (Fig. 3). This is surprising, given the importance of insularity in the global models and the role of geographic range size in classifying species' extinction risk[48]. This finding is likely because, relative to its variation at a global scale, land mass area shows limited variation within many regions. Similarly, invasive alien species were also of limited importance in some regions. This may be an artefact of the national measurement scale for this covariate, which could limit variation within some regions. Furthermore, the lack of congruence in variable importance may, in part, be attributable to spatial variation in sampling effort. Global data on species occurrence and threat levels are likely to be heterogeneous in both quality and precision. Threatened species richness may be elevated or depressed in proportion to total species richness in regions with incomplete sampling, potentially biasing results. Sampling effort is also likely to vary between taxonomic groups. The completed assessment for the distribution of reptiles is not currently available[49]. To test for an effect of spatial variation in sampling effort we repeated our analyses including data deficient (DD) species richness under several alternative assumptions of threat status. The distribution of DD species is geographically non-random, with greatest DD species richness in tropical regions[50]. These additional analyses produced qualitatively similar results that did not substantially differ from the main analysis of absolute threatened species richness (Supplementary Fig. 13), suggesting that our results are robust to spatial variation in sampling effort. Nevertheless, enhanced efforts to address gaps in biodiversity monitoring, such

as across tropical regions and for reptiles, will help to address any remaining uncertainties in regional variations in the drivers of imperillment[50].

**Responses of total and threatened species richness differ**. To explore the functional form of relationships between covariates and species richness patterns, we fitted separate models to both threatened and total global vertebrate species richness. In several cases, there were striking differences in the responses of total and threatened species richness to both environmental and human impact processes (Fig. 4). For amphibians, variation in elevation was positively related to threatened species richness, but negatively related to total species richness. Topographically diverse environments promote speciation and diversification through geographic isolation[26], yielding narrow-range endemic species, which are more susceptible to extinction owing to a lack of refuges from adverse environmental conditions. The decline in amphibian species richness contrasts with what would be expected from the mid-domain effect (i.e., the peak in species richness towards the centre of a shared geographic domain due to boundary constraints[51]), but has been reported previously[52]. It is likely caused by reductions in temperature associated with increasing elevation gradients that occur at too fine a spatial resolution to be reflected by our coarse resolution measure of annual mean temperature[53]. Threatened amphibian species richness showed a broadly linear positive relationship with long-term land use change, whilst total amphibian species richness remained relatively constant. Similarly, total reptile species richness remained relatively constant with long-term land use change. Threatened reptile species richness, however, showed a positive asymptotic relationship with long-term land use change (Fig. 4).

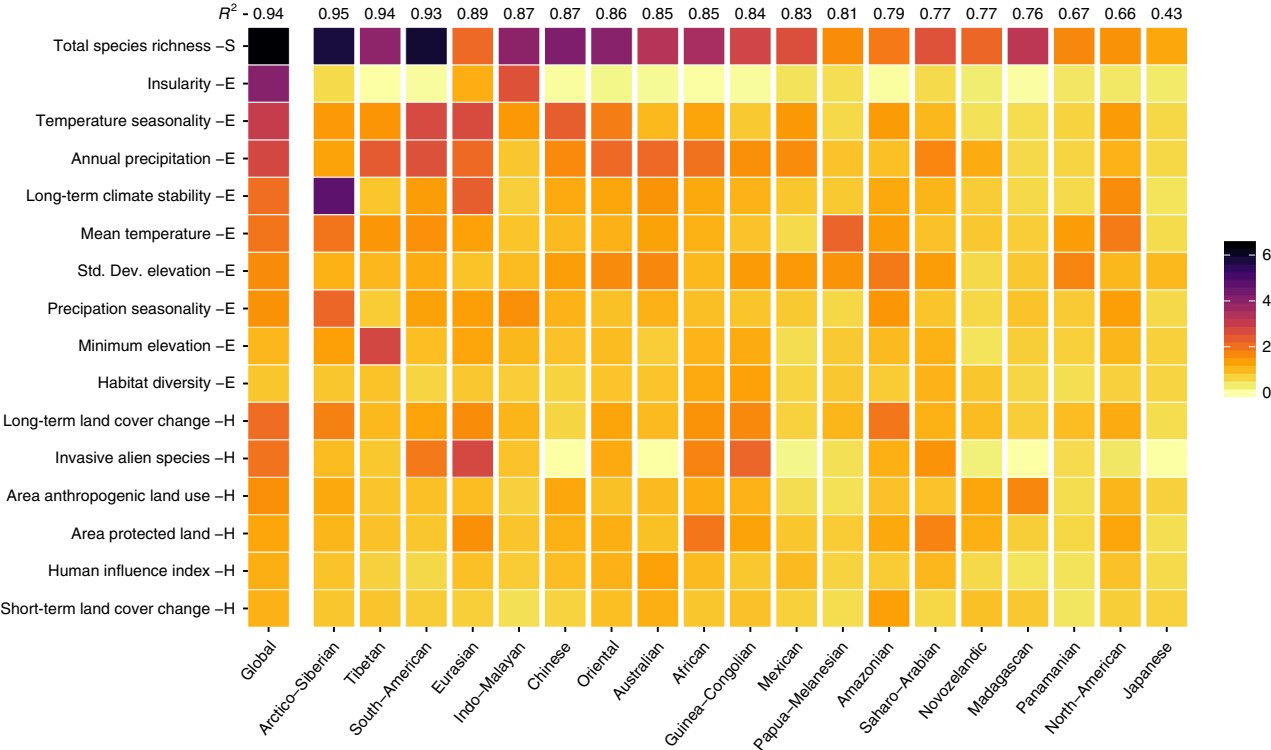

**Fig. 3 Importance of variables in predicting threatened vertebrate species richness in different zoogeographic regions.** Variables are grouped into broader classes, which are indicated by the capital letters on the side of the variable names: Total Species Richness (S), Environmental (E), and Human Impact (H) covariates. The mean importance of each variable from the models of threatened vertebrate species richness is indicated by the yellow (low importance) to black (high importance) colour gradient. Variables are ordered from top to bottom first by group, and then according to their importance in the global model of threatened vertebrate species richness. Average performance of each regional set of models, measured with $R^2$, is indicated at the top of each column, with regional models ranked by decreasing mean $R^2$. Heat maps of variable importance for the individual taxonomic groups can be found in Supplementary Figs. 5–8. Source data are provided as a Source Data file.

For both birds and mammals, the area of human dominated land uses showed a positive relationship with the number of threatened species in an area, whilst the relationship with total species richness had an intermediate maximum (Fig. 4). These relationships may indicate two processes at work. The initial positive relationship between species richness and human-dominated land uses may reflect how both respond positively to net primary productivity[32]. The subsequent reduction in total species richness and increase in threatened species richness may be a consequence of the associated increase in habitat loss and fragmentation in areas of extensive human-dominated land uses[54,55]. The relationship between insularity and mammal species richness also differed between the total and threatened species pools. Whilst the highest total mammal species richnesses were observed in areas with the greatest land mass, the highest threatened species richnesses were observed in the most insular areas. Islands are known to be centres for range-restricted species, more prone to extinction[30]. They are also, however, acknowledged for having a lower species richness than mainland regions, with species richness increasing with area[56,57].

**Implications for conservation planning.** Our global assessment of the drivers of threatened species richness reveals that those environmental characteristics that predispose species assemblages to threat may be more influential than threatening human processes for determining where concentrations of imperilled species occur. This finding does not suggest that threatening human activities are unimportant for driving species' extinctions; indeed, we shown that human impact variables are influential, and

particularly at regional scales. Rather, it suggests that some regions and taxonomic groups may be inherently more vulnerable than others to the effects of those threatening processes. This finding has important implications for conservation planning. Knowing the inherent vulnerability of a region can aid decisions regarding global conservation priorities and could form the basis for a biodiversity conservation roadmap[8]. For example, areas that are inherently more imperilled might also be at greater risk from the effects of threatening human processes, and could therefore be prioritised for stricter protective actions whilst mitigation—by remediation, for instance—may be more appropriate in less vulnerable areas. Furthermore, we uncover striking variation both between taxonomic groups and across space in the environmental mechanisms that are responsible for driving spatial gradients in threatened species richness. Any future coordinated global conservation strategy must consider the full range of scales at which the drivers of threat operate on biodiversity and the varying sensitivities of species to different threatening processes[7,8]. Whilst our results cannot supplant local-scale, species-specific assessments, they offer the information necessary to understand the processes acting at a global scale. This is critical, as this is the scale at which large, irreversible changes occur and at which many of the economic and political processes that ultimately drive extinction operate. By identifying the drivers of spatial gradients of threatened species richness, our approach has the potential to inform proactive conservation policy and action to minimise the effects of future environmental change. Combining our models with projections of environmental change would provide a baseline from which to predict how threatened species richness

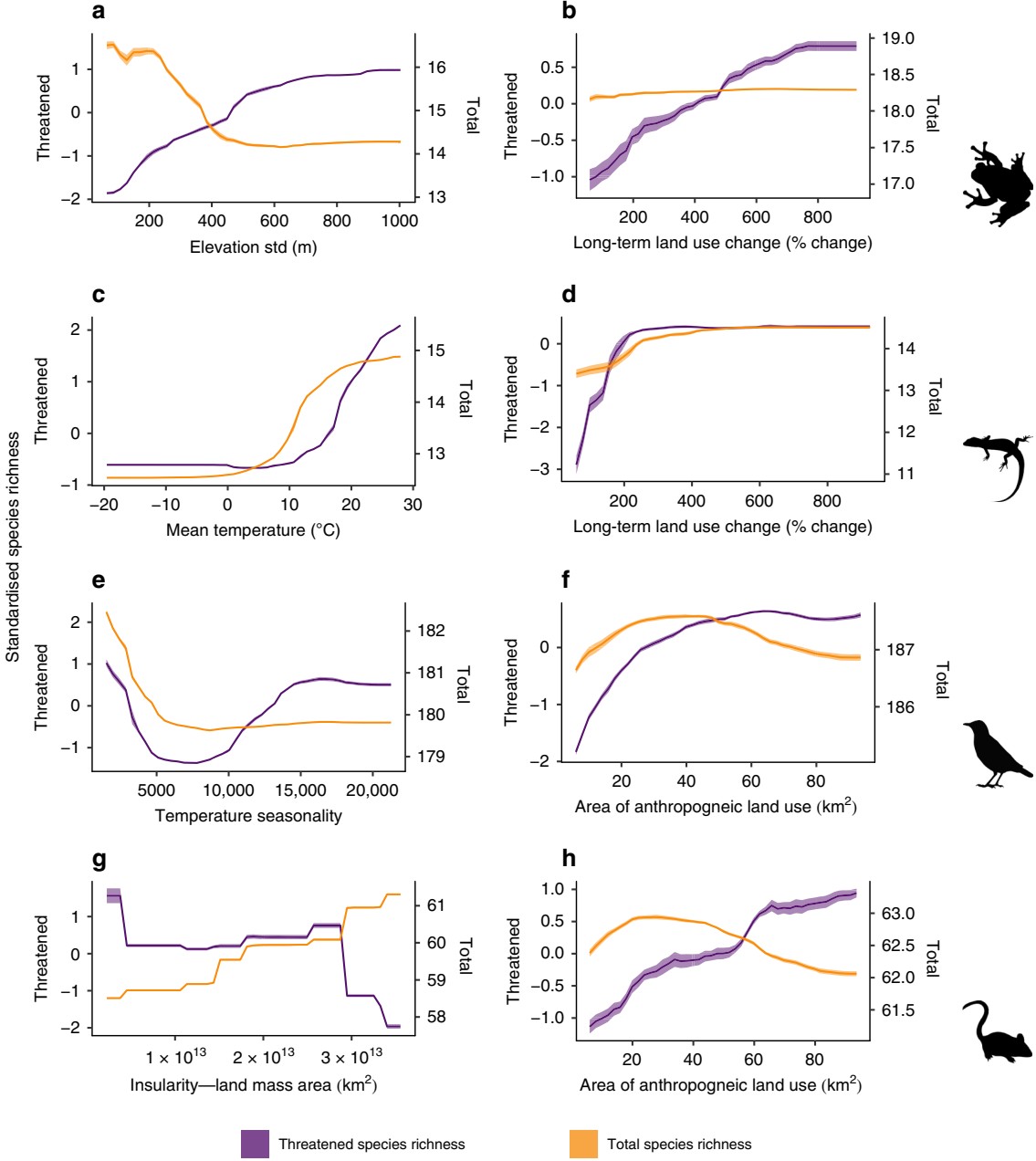

**Fig. 4 Relationships between a selection of the most important environmental and human impact covariates and global amphibian, reptile, bird, and mammal species richness.** Lines indicate the mean partial relationships between variables and threatened (purple) and total (orange) species richness from across 10 random forest models for amphibians (**a**, **b**), reptiles (**c**, **d**), birds (**e**, **f**) and mammals (**g**, **h**). Shaded areas indicate the standard deviation around the mean partial relationships. The x-axis is limited to the central 90% of a variable's range. Note the separate y-axes for threatened and total species richness. Plots of all variable relationships can be found in the Supplementary Figs. 10 and 11. Source data are provided as a Source Data file.

will respond to varying elements of global change and identify where concentrations of imperilment may occur in the future.

## Methods

**Species richness data**. Global distribution data for 6401 species of amphibians, 4450 reptiles and 5312 mammals were obtained from[49], whilst distribution data for 10,583 bird species were obtained from[58]. Data were available as spatial polygons of distribution boundaries, which were projected onto a Behrman equal-area projection and converted to a grid with a cell size of 0.5° at 30°N and 30°S latitude. Where a species' range polygon intersected with a grid cell, the species was treated as present within the entire cell. For each species, we only considered parts of the range where the species is designated as native, extant and either breeding or resident. Assessments of global extinction risk were taken from the IUCN[49]. Species richness was calculated as the total number of species present in a grid cell.

Threatened species richness was calculated as the total number of threatened species in a grid cell, where threatened species were those classified as either 'vulnerable', 'endangered' or 'critically endangered' by the IUCN. Total species richness and total threatened species richness were also calculated for the four individual taxonomic groups (amphibians, reptiles, birds and mammals, Supplementary Figs. 1 and 2).

**Environmental covariates**. Data on four bioclimatic variables were obtained from WorldClim (Version 1.4[59]); including annual mean temperature, temperature seasonality, annual precipitation and precipitation seasonality. Data were available on a 4.5 km grid for the 30-year period between 1960 and 1990. We derived minimum elevation and standard deviation of elevation using the Global Multi-resolution Terrain Elevation Data 2010 (GMTED2010[60]), available at a resolution of 7.5 arc-seconds. Land cover data were derived from GlobCover version 2.3 2009 (http://due.esrin.esa.int/page_globcover.php), which are available as a raster with

23 land cover classes at a resolution of 300 m. These data were used to estimate the diversity of land cover classes using the Shannon information index[61]. The total area (km[2]) of the land mass of which a grid cell is part was calculated to provide a measure of insularity, or how isolated a grid cell is from other land masses. Smaller and more isolated landmasses (i.e., islands) have a smaller value of insularity, whilst grid cells on continental landmass have a larger value. As range size is used as one of the criteria for classifying species' extinction risk, the inclusion of insularity may introduce circularity into these analyses. However, when compared with mainland areas, islands are known for their higher levels of endemic, range-restricted species that are potentially more vulnerable to imperilment[30]. By including insularity, we can explore the role of islands in promoting concentrations of threatened species richness alongside the suite of other variables considered here. A measure of temporal climate stability since the last interglacial period (present to 125,000 years ago) was derived from palaeo-climatic data made available by the Bristol Research Initiative for the Global Environment (BRIDGE, http://www.bridge.bris.ac.uk/). Details of the calculation of the model used to derive the paleo-climate data are provided by[62,63]. Data on precipitation and temperature were sampled at 4,000 year intervals. For each temporal transition, the Euclidean distance was calculated between z-transformed temperature and precipitation in bivariate space. The mean Euclidean distance was used as a measure of long-term climate stability in a grid cell, with smaller values indicating more stable climates[64].

**Human influence covariates**. The total area of land within each grid cell classified as intensively used by humans, as agricultural crop land or as urbanised areas, was obtained from GlobCover 2009[65]. As the occurrence of intensively used lands can manifest in areas with minimal human settlement[66], we also included the Human Influence Index (HII, http://sedac.ciesin.columbia.edu/wildareas/) as an additional measure of human impact. As a proxy for protection status, the total area of land falling in the International Union for Conservation of Nature (IUCN) protected area categories I–VI was calculated using the World Database on Protected Areas (WDPA; https://www.protectedplanet.net/). Short-term land cover change was quantified using global land cover data obtained from the European Space Agency Climate Change Initiative (ESA CCI, https://www.esa-landcover-cci.org/?q = node/1). These data were used to calculate the percentage change in the area of human influenced land cover classes between 1992 and 2015. To quantify long-term land cover change, the percentage change in the area of cropland between 1700 and 1992 was estimated using data derived from[67]. Data on the number of invasive alien species in each country were obtained from[68]. All human impact and environmental covariate data were aggregated onto the same 0.5° equal area grid used for the species distribution data. Further information on the derivation of explanatory variables can be found in the Supplementary Methods.

**Zoogeographic regions**. Zoogeographic regions were the 20 regions identified in[29]. These data were available as range polygons, which were also intersected with the same grid used for the species distribution data. Grid cells were classed as the region with the greatest coverage within the cell. As fewer than 10 cells were classified as belonging to the Polynesian region, this region was excluded from all analyses.

**Statistical models**. Random Forests were used to assess the potential of the covariates to explain the distribution of threatened species richness. A machine learning technique, random forests are a bootstrapped-based classification and regression tree method that are robust to overfitting, and are recognised as producing good predictive models[69]. They make fewer assumptions than correlative approaches about the distributions of predictor and response variables. Tests of collinearity between predictor variables revealed that only temperature seasonality and mean annual temperature had an absolute correlation of greater than 0.7 ($\rho = 0.85$, Supplementary Fig. 4). Models were fitted using the 'randomForest' package in R[70,71]. All analyses were carried out in R 3.3.1[71].

First, we modelled threatened species richness in relation to total species richness alone. Models were fitted at a global scale for the threatened species richness for the four individual taxonomic groups (amphibians, reptiles, birds and mammals) as well as the total threatened vertebrate species richness (i.e., the four taxonomic groups combined). Second, we modelled threatened species richness in relation to the suite of environmental and human impact covariates detailed above. These models also included total species richness as a covariate, to account for the size of the species pool. This approach enables both the independent effect of the total species richness to be controlled for, whilst also allowing for potential interactions with other covariates[72]. To explore the potential of taxonomic variation in the drivers of threatened species richness we fitted models both for the total threatened vertebrate species richness as well as for the threatened species richness for the four individual taxonomic groups. To explore spatial variation in the drivers of threatened species richness, we then fitted separate models for the 19 individual zoogeographic regions. Again, we did this for both total threatened vertebrate species richness and for the four individual taxonomic groups. Finally, to verify the results from these models, we also modelled the residual threatened species richness from the first set of models where threated species richness was modelled in relation to total species richness alone (Fig. 1). This third set of models used the same environmental and human impact covariates as before but omitted

total species richness. The results from these models were qualitatively very similar to those from the models of absolute threatened species richness and are reported in Supplementary Fig. 12. The model fitting process was the same for all models, regardless of scale or response, and is described below.

To account for potential spatial autocorrelation[73], we utilised a blocking method[74], to split the data into ten sampling blocks to be used for cross-validation. The creation of each sampling block was based on ecoregions[75] (http://www.worldwildlife.org/science/data). Within our study area, each non-contiguous area of an ecoregion was classified as a separate sampling unit. These sampling units were then grouped into ten blocks so that the mean value of each environmental variable was similar across all blocks, but each block covered the full range of environmental variables within the area of study[76]. As all blocks cover a similar range of environmental data, this method ensures that a similar range of data was used for both calibrating and testing models, whilst also ensuring that the calibration and testing data are spatially segregated[77]. This method performs well at a large spatial scale, by minimising the influence of spatial autocorrelation whilst allowing models to capture complex spatial processes[74].

We used ten-fold cross-validation to find the optimal values of both the number of trees (nt) and the number of predictors (m) used to build each regression tree, which form the random forests. We initially fitted a random forest with nt set to 1000 and m set to 1. The initial model was calibrated on 9 of the 10 sampling blocks and performance was evaluated on the omitted sampling block. We used the coefficient of determination ($R^2$) to assess model performance on the evaluation data. A further 500 trees were then added to the model and the $R^2$ was recalculated on the evaluation data. If the additional trees improved model performance by more than 1% they were accepted. This was repeated iteratively until model performance was not improved by additional trees. We then repeated this process with m set to 2 and 3. This process of fitting models with varying values of nt and m, was repeated ten times, omitting a different sampling block each time. The values of m and nt that maximised $R^2$ across the ten iterations were then used to fit the final set of models. Again, this used a ten-fold cross-validation approach, fitting the model to nine of the ten sampling blocks and evaluating performance on the omitted block using $R^2$, omitting a different sampling block each time. This resulted in ten random forest models of threatened vertebrate species richness. A full summary of final model parameters and performance can be found in Supplementary Tables 1 and 2.

**Calculating variable importance**. The importance of individual variables was calculated using a randomisation approach. We initially assessed model performance using mean squared error (MSE) on predictions made to the full data set. We then randomised a variable, made new predictions and recalculated MSE. The importance of the variable in the model was then calculated using Eq. 1:

$$\text{VI} = \sqrt{\frac{\text{MSE}_{rand} - \text{MSE}_{obs}}{\text{MSE}_{obs}}} \tag{1}$$

where VI is variable importance, $\text{MSE}_{rand}$ is the mean squared error from the predictions using the randomised variables, and $\text{MSE}_{obs}$ is the mean squared error from the predictions with the variable as observed. We repeated the process 1000 times for each variable in the model and reported the mean VI.

To investigate the potential for spatial inconsistencies in the drivers of threatened vertebrate species richness, we repeated the above analyses for individual zoogeographic regions. Both global and zoogeographic regional level analyses were repeated for the four individual taxonomic groups to assess cross-taxon congruence in the drivers of threatened species richness. Owing to the small sample size of some sampling blocks for the individual regions, some models lacked explanatory power (i.e., $R^2 < 0.25$[78]). These models were excluded from calculations of variable importance and any further analyses. All other models showed medium to excellent performance (Supplementary Table 1). We used repeated measures ANOVAs to identify differences in the explanatory power of environmental and human impact covariates. For this we aggregated the variables into two broad categories (and total species richness), and then took the mean importance across all variables within each category[79].

**Variable relationships**. To explore the functional form of relationships between individual variables and species richness patterns, we fitted separate models to both threatened and total species richness, using the model fitting procedure outlined above. This was performed for the individual taxonomic groups at a global scale, using the same set of predictor variables, barring total species richness. Predictions were then made to a data set where all but the focal variable were held at their mean (or modal) value. This was repeated for each variable for both responses (total and threatened species richness) for the four taxonomic groups. Here, we present a selection of the most important environmental and human impact variables from the taxa specific global models of threatened species richness, for which the response of total species and threatened species richness differs, while also being ranked in the top three of variable importance for each of the environmental and human impact categories. Plots of the relationships between all variables and both threatened and total species richness, along with the measures of variable importance for these additional models, can be found in Supplementary Figs. 9, 10 and 11.

**Accounting for data deficient species richness**. There is some evidence that data deficient (DD) species are far more likely to be threatened than categorised species[80]. Furthermore, there is evidence that the distribution of DD species is spatially non-random, with concentrations of DD species occurring throughout low latitude, tropical regions[50] (Supplementary Fig. 3). To explore the sensitivity of our results to the inclusion of DD species, we refitted our models, using the process outlined above, but including DD species under several different assumptions of threat status. For this, we randomly classified 0, 50 and 100% of DD species that occur in each grid cell as threatened, and then included these species in our calculations of total and threatened species richness, accordingly. The different assumptions of threat status for the DD species produced qualitatively similar results that did not substantially differ from the main analysis of absolute threatened species richness (Supplementary Fig. 13).

**Reporting summary**. Further information on research design is available in the Nature Research Reporting Summary linked to this article.

## Data availability

The species richness data generated and analysed during this study are included in the Source Data files. The other datasets utilised in this study are derived from published sources, cited in the methods section. The source data underlying Figs. 1–4, Supplementary Figs. 1–13, Tables 1–2, and Supplementary Tables 1–2 are provided as a Source Data file.

## Code availability

Code to carry out analyses is publicly available on https://github.com/christinehoward399/Global-Rarity.

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

## Acknowledgements

This research was supported by the Inventory, Monitoring & Assessment research staff of the US Forest Service through International Joint Venture Agreement 14-JV-11221636-114 between the Rocky Mountain Research Station and Durham University. We thank BirdLife International for providing bird species distribution data and Stuart H.M. Butchart at Birdlife International for commenting on the manuscript. We also thank Paul J. Valdes from the Bristol Research Initiative for the Global Environment (BRIDGE, http://www.bridge.bris.ac.uk/) for providing paleo-climatic data and commenting on the manuscript.

## Author contributions

C.H., C.H.F. and P.A.S. designed the study. C.H. performed the analysis. C.H., C.H.F., and P.A.S. wrote the manuscript.

## Competing interests

The authors declare no competing interests.
