## [Peer Review File · Nature Communications]

Reviewers' comments:

Reviewer #1 (Remarks to the Author):

This is an interesting analysis of the drivers of threatened species richness. The authors use the distribution and extinction risk (Red List status) of tetrapods to test the role of different environmental and human impact drivers. They complete comprehensive analyses but I found the interpretation of results a bit confusing and have some queries about the methods.

First, in line 78 "Furthermore, and contrary to previous reports 37, the limited influence of anthropogenic variables suggest that this result is unrelated to enhanced human impacts in areas of greater net primary productivity 38–40" Human activities and species richness are both influenced by environmental variables. I am not convinced the analyses used can rule out the possibility that environmental variables may be acting as proxies of evolutionary/ecological drivers of extinction risk AND of human pressure, in which case the importance of human pressure could be underplayed. Random forest analyses can be affected by correlations among variables (Nicodemus, K.K. et al. The behaviour of random forest permutation-based variable importance measures under predictor correlation. *BMC Bioinformatics* 11, 110 (2010)). So environmental variables may be taking a more dominant role due to their dual effect.

Second, the authors initially claim human impacts play a minor role but in their analyses by realms they find something else, line 116-118 "In areas where threatened species richness is greater than expected given the total species richness (Figure 1a), the most influential variables were often related to anthropogenic threatening processes." A recent study (Polaina, et al (2018) *Global Ecology and Biogeography* 27: 647-657 DOI: 10.1111/geb.12725) has shown the relationship between threatened mammalian richness and human land is not globally unique or linear, which means that when analysing global patterns, the distinct relationships can be muddled and appear as no effect. I think the findings of the ms are more nuance (and arguably interesting) than the current ms suggests.

Figure 4 presents a series of challenges for interpretations. First, I am also unclear why these particular variables were plotted as they include variables with high but also low importance. For example, habitat diversity is shown here but we can see it is the least informative (important) variable in all cases in figure 2, including in mammals. The functional relationships of variables with limited importance is not particularly meaningful (as these variables are not useful to predict the response). On the other hand, insularity is very important for all and for mammals, but that relationship is only shown in the supplementary materials. Second, I found it difficult to interpret figure 4 without the results of variable importance for the total species richness model as again the relationship is not very relevant for low importance variables. In order to assess whether the relationships for total and threatened richness are meaningfully different I would like to see something like figure 2 for the total species richness model. Third, I appreciate the value of standardizing the response but this also means we cannot easily assess the magnitude of the

biological effects. For example, in line 139 the authors write “For both amphibians and reptiles, areas with more invasive alien species had more threatened species, but fewer species overall” how many fewer species overall or more are we talking about? Is it one more species or 10? (this brings up a detail missing from the methods about how origin was considering in the distribution data from the IUCN which I describe below). As a minor detail area of anthropogenic land use is misspelled in figure 4.

Species can be classified as threatened based on having small range areas, which are not rare in insular species, so could this finding be just an issue of circularity? Insular distribution used to define threatened status and thus correlated with it when later tested.

The methods and supplementary material separate Environmental Covariates and Anthropogenic Influence variables, but then figure 2 has four different categories, it would be useful to have consistent groupings.

The distribution data for reptiles does not reflect all reptilian biodiversity (the assessment is not yet completed, or better said, the completed assessment is not available yet), using data for a potentially spatially biased dataset on reptiles could be a problem here and should be at minimum discussed.

Line 184, Polygons also include native and introduced areas, where non-native ranges considered in the dataset?

I am not sure values of invasive species per country are a good source of information for what happens within grid cells of 0.5 degrees. I realize there is no better spatially-explicit dataset of invasive species, but the limitation should be discussed.

In figure 1 the class figures are difficult to see. In amphibians, which is one of the groups with more threatened species it is interesting that there seem to be no geographic patterns in threat (no colours in the map), Do the authors have any insights into why this may be?

Reviewer #2 (Remarks to the Author):

This study addresses a relatively straightforward yet surprisingly understudied question: are geographic patterns of variation in richness of threatened vertebrate species driven more strongly by environmental features or by anthropogenic land use change? The authors use the full suite of global vertebrate geographic range datasets now available to provide strong evidence in support of the former. They do a good, thorough and well thought out job of the analysis. For example, it was good to see that they tested for and considered spatial autocorrelation, whereas most users of the Random Forest method dismiss this as supposedly unnecessary (even though there is evidence from the statistical literature to the contrary, eg Sinha et al 2019, *Computers, Environment & Urban Systems* 75: 132-145).

There are a few areas in which I think the authors are a little too ready to accept their results at face value without acknowledgment of possible biasing factors. One issue is inadequate occurrence records – many tropical regions such as New Guinea are notoriously poorly sampled for amphibians, reptiles, and even mammals. This could conceivably either elevate threatened species richness relative to total richness (if undersampling leads to more species being incorrectly considered rare), or depress threatened richness (if undersampling leads to fewer genuinely rare species being recorded).

Another issue is Data Deficient species, which make up a substantial proportion of all vertebrate taxa except birds, but which weren't considered by the authors in these analyses (as far as I could tell). At least for mammals there is evidence that DD species are far more likely to be threatened than categorized species (eg Bland et al 2015 *Cons Biol* 29: 250-259; Jetz & Freckleton 2015 *Phil Trans B* 370: 1662). Because the distribution of DD species is very likely geographically non-random (mostly in the tropics), their inclusion in the models could change the geographic patterns of threatened species richness and the associations with external factors. The authors should consider running additional analyses that include DD species under several alternative assumptions of threat status.

I couldn't see the full model outputs anywhere. These should be provided in the SI, and at least some of the major model outputs (eg Table S1 & S2) should be in the main text.

Minor issues:

Line 84: should also cite Davies et al *PLoS Biol.* , 9, e1000620

Line 97-100: should also cite recent Science paper by Scheele et al. Isn't this also consistent with what is said on the previous page about diversification and extinction risk being linked? Here there seems to be a different explanation offered for the same pattern.

Line 116-118: I'd guess that habitat loss is important for areas with greater than expected threatened richness because these are areas with many of the narrow-range endemics you mention

earlier – these are especially sensitive to habitat loss. Likewise, total modified area is imp't for Madagascar most likely because most of Madagascar's original habitat has been lost.

Line 133: Elevational variation was negatively related to total amphibian richness – really?

Line 235: How strongly correlated are threatened and total species richness? Are there likely to be collinearity issues?

Marcel Cardillo

Reviewer #3 (Remarks to the Author):

This paper contrasts patterns of total species richness and threatened species richness, therefore identifying regions where species are inherently more vulnerable, and guiding conservation prioritisation. We (a PhD student and supervisor team; we got prior editorial approval for this) believe that this is a very well written paper and a useful contribution to the field of conservation biology.

However, we have two reasonably major general concerns that we think need to be addressed before publication. These are:

1. Is the small effect of anthropogenic disturbance in the results not simply the result of using 0.5 degree data? The issue we see here is that simply examining the percent cover of urban/ag per cell (or mean Last of the Wild, etc) at this resolution is likely to considerably weaken the signal of human disturbance, as where within such a large grid cell this disturbance is occurring relative to where the biodiversity is occurring will have a major impact on how much the disturbance matters. We recognize there is not much that can be done about this, but this limitation – and hence the limitation of the inferences made about this pressure - need to be very clearly spelled out in the Discussion. We also think that historical land use change between 1700 and 1992 is not the only useful predictor of pressure, as for many species' threats may have accelerated since then. Why not also use the ESA CC data (1992 to 2015 harmonized land cover data at 300 m resolution) to get an idea of recent trajectories of change per degree cell?

2. Issues with the Random Forest (RF) models. More seriously, we have several concerns about the RF models that this manuscript is based on. First (this is a fairly minor point, but important to deal with), saying you have an R2 in the main results (and Table S1) of 0.94 is misleading. Looking at the methods in detail, it seems you've actually done post-hoc cross-validated correlations of the RF models built on 9 of your 'blocks' with the predicted values from the remaining independent 'block'. This is decidedly non-standard; a quick reading of the main text would suggest to most readers that

you've done some sort of linear global model and gotten these very high R2 values. This needs better explanation. Secondly, and more importantly, we have concerns about whether RF properly corrects for total species richness by including it as a covariate. This is because RF works by removing a variable and then seeing how well the trees perform after it's been removed for all combinations of the other variables, over many trees. This means that the relative importance value of each of the other variables is essentially based on trees that do – and do not – include species richness in them. As such, variable importance does not fully control for species richness. The authors do also include models that are based on the residuals from models of threatened species richness against species richness, but it's not clear what the initial model here actually is (is it just a RF with species richness and threatened species richness in it?). Using residuals is frowned upon in ecology (as the authors are aware; they cite Freckleton 2002) as biases the coefficients if species richness is correlated with the predictors, as will clearly be the case. As such, this extra check does not necessarily deal with the issue we outlined earlier about controlling for species richness in RF. As such, we suggest that you do additional analyses – at least globally for each group – using standard macroecological generalized linear modelling approaches that account for SAR with species richness as a predictor. Such models will give unbiased estimates of the non-species richness predictors after controlling for species richness (see Freckleton 2002).

3. Finally, we're not sure that the 'take 9 blocks and predict block 10' approach is a very good test of predictive power. If all 10 blocks are within the same environmental space, it's hardly surprising that you have very high R2 values. Why not simply take the average pseudo R2 values from RF models run for various combinations of all 10 blocks? RF does internal cross-validation, so that way you still correct for SAC, but can use standard RF reporting outputs.

We also have some specific, fairly minor suggestions, outlined below:

Line edits (lines)

Title

1 – We would potentially suggest changing 'imperilled' in the title to 'threatened', as threatened has a clearer definition and is more widely understood. Unless imperilment is used to refer to the difference between total species richness and threatened species richness. If so, this should be made explicit in the abstract and the main text.

Careful use of 'imperilment' vs 'threatened species' is required throughout the manuscript it would be best not to use them interchangeably.

Abstract

14 – It might be informative to suggest what types of environmental factors and habitat characteristics are typically more important than anthropogenic impacts for explaining spatial variation in numbers of threatened species. E.g., environmental factors, such as temperature seasonality and precipitation, and habitat characteristics, such as insularity, are typically more important ...

The findings in the abstract are currently very generally and more specific details would be useful for the readers.

15 – Personally, we prefer the use of the term ‘human’ instead of ‘anthropogenic’, as we think using the word human is more readable and the word anthropogenic often leads to more detached thinking that reduces the feeling of our collective human responsibility. Up to the authors though.

16 – We would mention the focal taxa in the abstract for clarity, so instead of saying ‘these relationships vary between taxa’ you could say ‘these relationships vary between terrestrial vertebrate groups’ or similar.

Introduction

21 – The spaces before references should be removed throughout the manuscript.

43 – It could be easier for the reader if the order of ‘predisposing environmental conditions’ and ‘anthropogenic threatening processes’ is consistent throughout the manuscript, including the order they are introduced (line 37). For us, it makes most sense to describe or refer to the predisposing environmental conditions first and the extrinsic threatening processes second.

56 – Consistent use of environmental/habitat/energy/anthropogenic variables is needed. Which variables count as environmental, which as habitat related? In the abstract (line 14) they are described as environmental, habitat and anthropogenic, whereas in the Results and Discussion (line 67) they are referred to as habitat, energy and anthropogenic. Also for Figure 2 and Figure 3.

Results and Discussion

73 – For us, Supplementary Figure 3 does not equivocally confirm the idea that island regions support a higher number of endemic, extinction-prone species than continental land masses, as insularity shows a bimodal distribution. Therefore, we would suggest dropping reference to Supplementary Figure 3 here.

75 – Check use of superscript for 1st, 2nd, 3rd, etc. Currently some are superscript and some are not.

93 – What is the word ‘significantly’ here based on? If it’s not based on a statistical test the term ‘significantly’ could be dropped.

106 – We would suggest defining ‘vagile’ or using a simpler expression, as the readership of this journal is broad.

Methods

195 – Need a reference for the source of the elevation data.

275 – It would be useful to state the values of m and n_t that were actually used to fit the final set of models.

291 – Consistency is needed for the use of ‘zoogeographic’ vs ‘biogeographic’ realms. We assume that the zoogeographic realms referred to in the statistical analyses (line 291) are the biogeographic realms described previously (line 223). Also for Figure 3.

294 – Which models lacked explanatory power, which regions?

478 – Figure 2, duplication of information for types of variables. Either drop letters or drop colours. If letters are kept they need to be described in the Figure caption.

487 – Figure 3, for me, decreasing mean R^2 would make more sense, as the best explained models would appear first.

487 – Figure 3, why does the model for the Japanese Biogeographic/Zoogeographic realm have low explained variance? Might be nice to suggest why this might be in the main text.

495 – Figure 4, why were the selected variables selected. A short description of the selection criteria would be informative.

Supplementary

Supplementary Table 2 – There is no description of this ANOVA in the methods. Needs to be described in the main or Supplementary methods

Supplementary Figure 3 – y-axis label is misspelt, richness is spelt as 'richnes'

Response to Reviewers: NCOMMS-19-15718A / NCOMMS-19-15718-T

Reviewer Comment	Response
Reviewer 1	
First, in line 78 “Furthermore, and contrary to previous reports 37, the limited influence of anthropogenic variables suggest that this result is unrelated to enhanced human impacts in areas of greater net primary productivity 38–40” Human activities and species richness are both influenced by environmental variables. I am not convinced the analyses used can rule out the possibility that environmental variables may be acting as proxies of evolutionary/ecological drivers of extinction risk AND of human pressure, in which case the importance of human pressure could be underplayed. Random forest analyses can be affected by correlations among variables (Nicodemus, K.K. et al. The behaviour of random forest permutation-based variable importance measures under predictor correlation. BMC Bioinformatics 11, 110 (2010)). So environmental variables may be taking a more dominant role due to their dual effect.	This is an important point. Random forests are reported to be robust to correlated variables¹, but as the reviewer highlights there is evidence that correlated variables can impact variable importance metrics². We have tested for collinearity between predictor variables. Only temperature seasonality and mean annual temperature had an absolute correlation greater than 0.7 ($r = -0.85$). We believe it unlikely that environmental variables are acting as proxies for both ecological drivers and human impacts, as all combinations of environmental covariates and human impact variables had an absolute correlation of less than 0.5. We have now added text to the methods to detail our tests for collinearity (Lines 257-260) and a correlation plot in the supplementary materials (Supplementary Figure S4).
Second, the authors initially claim human impacts play a minor role but in their analyses by realms they find something else, line 116-118 “In areas where threatened species richness is greater than expected given the total species richness (Figure 1a), the most influential variables were often related to anthropogenic threatening processes.” A recent study (Polaina, et al (2018) Global Ecology and Biogeography 27: 647-657 DOI: 10.1111/geb.12725) has shown the relationship between threatened mammalian richness and human land is no globally unique or linear, which means that when analysing global patterns, the distinct relationships can be muddled and appear as no effect. I think the findings of the ms are more nuance (and arguably interesting) than the current ms suggests.	We have now made additions to the text to emphasise both that the low importance of human impact covariates is a global-scale specific result (Line 70) and that our findings vary substantially when considered at a regional scale (Lines 106-108).
Figure 4 presents a series of challenges for interpretations. First, I am also unclear why these particular variables were plotted as they include variables with high but also low important. For example, habitat diversity is shown here but we can see it is the least informative (important) variable in all cases in figure 2, including in mammals. The functional relationships of variables with limited importance is not particularly meaningful (as these variables are not useful to predict the response). On	Apologies: we originally selected variables with interesting contrasts between their relationship with total and threatened species richness. We have now made sure that all variables presented in Figure 4 are of high importance in the taxa-specific threatened species richness models. The relationships for all variables are presented in Supplementary Figures S10 and S11 and we have amended the text accordingly (Lines 335-339). We now use separate axes for the total and threatened species richness relationships so

the other hand, insularity is very important for all and for mammals, but that relationship is only shown in the supplementary materials. Second, I found it difficult to interpret figure 4 without the results of variable importance for the total species richness model as again the relationship is not very relevant for low importance variables. In order to assess whether the relationships for total and threatened richness are meaningfully different I would like to see something like figure 2 for the total species richness model. Third, I appreciate the value of standardizing the response but this also means we cannot easily assess the magnitude of the biological effects. For example, in line 139 the authors write “For both amphibians and reptiles, areas with more invasive alien species had more threatened species, but fewer species overall” how many fewer species overall or more are we talking about? Is it one more species or 10? (this brings up a detail missing from the methods about how origin was considering in the distribution data from the IUCN which I describe below). As a minor detail area of anthropogenic land use is misspelled in figure 4.	that the magnitude of the effects can be displayed (Figure 4). We have also included a figure displaying the variable importance from these additional sets of models so that it is easier to contrast the importance of variables in models of total and threatened species richness (Supplementary Figure S9). Apologies, the spelling mistake has now been corrected.
Species can be classified as threatened based on having small range areas, which are not rare in insular species, so could this finding be just an issue of circularity? Insular distribution used to define threatened status and thus correlated with it when later tested.	Yes, there is the potential for circularity between insularity and threatened species richness, which we now discuss in lines 128-129. We feel, however, that it would be remiss not to include this as a predictor. Given that islands have been shown to be hotspots of threatened species richness³, we feel that it is important to understand the relative role of insularity in determining where concentrations of threatened species occur alongside the suite of other variables in our models.
The methods and supplementary material separate Environmental Covariates and Anthropogenic Influence variables, but then figure 2 has four different categories, it would be useful to have consistent groupings.	We agree. We have now grouped the variables into ‘Environmental’ and ‘Human Impact’ covariates and use these terms consistently throughout the manuscript. The use of ‘Human Impact’ is in response to a point made by reviewer 3.
The distribution data for reptiles does not reflect all reptilian biodiversity (the assessment is not yet completed, or better said, the completed assessment is not available yet), using data for a potentially spatially biased dataset on reptiles could be a problem here and should be at minimum discussed.	We now discuss this point in lines 136-138.
Line 184, polygons also include native and introduced areas, where non-native ranges considered in the dataset?	No, only native species ranges were included. We have now made this clear in the methods (Line 206).

I am not sure values of invasive species per country are a good source of information for what happens within grid cells of 0.5 degrees. I realize there is no better spatially-explicit dataset of invasive species, but the limitation should be discussed.	We now discuss this point in lines 130-132.
In figure 1 the class figures are difficult to see. In amphibians, which is one of the groups with more threatened species it is interesting that there seem to be no geographic patterns in threat (no colours in the map), Do the authors have any insights into why this may be?	The apparent lack of geographic patterns in amphibian species richness was a result of using the same colour scale for all panels of Figure 1. We have now used separate colour scales for each panel, which we feel improves the figure.
Reviewer 2:	
There are a few areas in which I think the authors are a little too ready to accept their results at face value without acknowledgment of possible biasing factors. One issue is inadequate occurrence records – many tropical regions such as New Guinea are notoriously poorly sampled for amphibians, reptiles, and even mammals. This could conceivably either elevate threatened species richness relative to total richness (if undersampling leads to more species being incorrectly considered rare), or depress threatened richness (if undersampling leads to fewer genuinely rare species being recorded).	This is a good point, the implications of which we now discuss on lines 135-146. To explore the potential impacts that spatial variation in sampling effort may have on our results, we re-ran our models of absolute threatened species richness including data deficient (DD) species richness as a predictor. This additional analysis assumed that DD species can act as a proxy for sampling effort, particularly given the tendency for increased DD species richness in the tropics where sampling effort may be assumed to be lowest (Collen et al., 2008, Supplementary Figure S3). The inclusion of DD species richness did not substantially alter the results; see Figure R1 at the end of this response.
Another issue is Data Deficient species, which make up a substantial proportion of all vertebrate taxa except birds, but which weren't considered by the authors in these analyses (as far as I could tell). At least for mammals there is evidence that DD species are far more likely to be threatened than categorized species (eg Bland et al 2015 Cons Biol 29: 250-259; Jetz & Freckleton 2015 Phil Trans B 370: 1662). Because the distribution of DD species is very likely geographically non-random (mostly in the tropics), their inclusion in the models could change the geographic patterns of threatened species richness and the associations with external factors. The authors should consider running additional analyses that include DD species under several alternative assumptions of threat status.	As suggested by the reviewer, we have run a set of additional analyses where we include DD species under different assumptions of threat status (lines 340-349). The results from these additional analyses did not substantially differ from the main results and are now included in the supplementary materials (Figure S13) and discussed in Lines 140-146.
I couldn't see the full model outputs anywhere. These should be provided in the SI, and at least some of the major model outputs (eg Table S1 & S2) should be in the main text.	A full summary of model outputs has now been included in the supplementary material (Supplementary Tables S1 and S2) and we have now included an overall summary of model performance

	in the main text (Table 1).
Line 84: should also cite Davies et al PLoS Biol. , 9, e1000620	Citation included (Line 75).
Line 97-100: should also cite recent Science paper by Scheele et al. Isn't this also consistent with what is said on the previous page about diversification and extinction risk being linked? Here there seems to be a different explanation offered for the same pattern.	Citation included (Line 93). We have also amended the text to maintain consistency (Lines 88-89).
Line 116-118: I'd guess that habitat loss is important for areas with greater than expected threatened richness because these are areas with many of the narrow-range endemics you mention earlier – these are especially sensitive to habitat loss. Likewise, total modified area is imp't for Madagascar most likely because most of Madagascar's original habitat has been lost.	This is a good point which we have now included in our discussion (Lines 113-114).
Line 133: Elevational variation was negatively related to total amphibian richness – really?	We believe this is due to variations in temperature that happen over elevational gradients that are at too fine a spatial scale to be reflected in our coarse measure of mean annual temperature. We have now discussed this in lines 154-159.
Line 235: How strongly correlated are threatened and total species richness? Are there likely to be collinearity issues?	There is some correlation between threatened and total species richness (mean $R^2 = 0.68$, S.D. ± 0.23). This is expected given that total species richness dictates maximum potential threatened species richness. This is why we account for total species richness in our models of threatened species richness. As we do not use both total and threatened species richness as predictor variables in the same model, this collinearity should not be an issue.
Reviewer 3:	
Is the small effect of anthropogenic disturbance in the results not simply the result of using 0.5 degree data? The issue we see here is that simply examining the percent cover of urban/ag per cell (or mean Last of the Wild, etc) at this resolution is likely to considerably weaken the signal of human disturbance, as where within such a large grid cell this disturbance is occurring relative to where the biodiversity is occurring will have a major impact on how much the disturbance matters. We recognize there is not much that can be done about this, but this limitation – and hence the limitation of the inferences made about this pressure - need to be very clearly spelled out in the Discussion.	We have now discussed how the resolution of our analyses may impact our results in Lines 117-124.
We also think that historical land use change between 1700 and 1992 is not the only useful predictor of pressure, as for many species' threats may have accelerated since then. Why not also use the ESA CC	This is a good suggestion. We have now included this measure of short-term land use change as a predictor variable in all models that include environmental and human impact covariates (Lines 238 - 241, Figures 2

data (1992 to 2015 harmonized land cover data at 300 m resolution) to get an idea of recent trajectories of change per degree cell?	and 3, Supplementary Materials).
More seriously, we have several concerns about the RF models that this manuscript is based on. First (this is a fairly minor point, but important to deal with), saying you have an R2 in the main results (and Table S1) of 0.94 is misleading. Looking at the methods in detail, it seems you've actually done post-hoc cross-validated correlations of the RF models built on 9 of your 'blocks' with the predicted values from the remaining independent 'block'. This is decidedly non-standard; a quick reading of the main text would suggest to most readers that you've done some sort of linear global model and gotten these very high R2 values. This needs better explanation.	Testing model performance on a semi-independent subset of data withheld from model training is a common approach. This is particularly true for species distribution models⁵⁻⁸, with which our models of species richness share many similarities. If we were to both fit our model and test the performance on the entire global dataset, we would get a much higher overall value of R². This is because the calibration and testing data would come from completely overlapping regions. Our approach to testing model predictive performance using data withheld from model fitting provides more conservative measure of fit⁸. We have now removed the inline reference to model performance (Line 61), to minimise risk of misinterpretation by a reader. Performance measures for all models are now reported in Supplementary Table S1 and S2. We have also clarified how model performance is calculated using the coefficient of determination in the methods (Lines 297-298).
Secondly, and more importantly, we have concerns about whether RF properly corrects for total species richness by including it as a covariate. This is because RF works by removing a variable and then seeing how well the trees perform after it's been removed for all combinations of the other variables, over many trees. This means that the relative importance value of each of the other variables is essentially based on trees that do – and do not – include species richness in them. As such, variable importance does not fully control for species richness. The authors do also include models that are based on the residuals from models of threatened species richness against species richness, but it's not clear what the initial model here actually is (is it just a RF with species richness and threatened species richness in it?). Using residuals is frowned upon in ecology (as the authors are aware; they cite Freckleton 2002) as biases the coefficients if species richness is correlated with the predictors, as will clearly be the case. As such, this extra check does not necessarily deal with the issue we outlined earlier about controlling for species richness in RF. As such, we suggest that you do additional analyses – at least globally for each group – using standard macroecological generalized linear modelling	We respectfully disagree with the reviewers' interpretation of variable importance in Random Forest models. The variable importance measures of Random Forest models are derived from a randomisation process. First, prediction accuracy is measured using the out-of-bag sample. The values for the variable are then randomly permuted, and prediction accuracy recalculated. The difference between the two measures is then calculated. This is repeated for all trees in the forest, over which a mean difference in performance is calculated and then normalised⁹. However, as variable importance is calculated after the regression trees have been fitted, this will not alter how an RF model accounts for a variable. The point where random forests may not properly control for total species richness is during the building of the trees. The variables used for each regression that form the individual regression trees, are randomly sampled from the covariate set (the number of which is determined by the m parameter). Given the number of trees that make up each forest (≥1000), and that the trees are left unpruned, it is

approaches that account for SAR with species richness as a predictor. Such models will give unbiased estimates of the non-species richness predictors after controlling for species richness (see Freckleton 2002).	highly likely that all variables within the model, including total species richness, are considered at multiple points throughout each forest, and hence are properly accounted for. As the reviewer has pointed out, we are aware that our check of model results using the residuals from the model of threatened species richness based on total species richness alone is not standard practice. However, given the global scale of these random forest models, and their non-linear nature, any comparison with a linear approach would not be meaningful. We have added text to clarify the structure of the model of residuals from a model of threatened species richness on total species richness alone (lines 276-278).
Finally, we're not sure that the 'take 9 blocks and predict block 10' approach is a very good test of predictive power. If all 10 blocks are within the same environmental space, it's hardly surprising that you have very high R2 values. Why not simply take the average pseudo R2 values from RF models run for various combinations of all 10 blocks? RF does internal cross-validation, so that way you still correct for SAC, but can use standard RF reporting outputs.	The out-of-bag data used by random forests for the internal cross validation is based on random k-fold sampling. This means that the calibration and training data sets are from overlapping regions. This means that the distances between calibration and testing data points tend to be much smaller for random k-fold CV than they are for a blocking approach. This spatial sorting bias in random k-fold CV tends to lead to over optimistic error estimates when compared with a blocking approach^{8,10-12}. We have included a table that demonstrates the over-estimation of model performance using the standard RF reporting outputs at the end of this response (Table R1). As you will see, the approach we have used leads to conservative estimates of model fit. Furthermore, our blocking approach also ensures a more robust consideration of spatial autocorrelation than that provided by a random forest's internal cross-validation (which evidence suggests is warranted¹³).
We would potentially suggest changing 'imperilled' in the title to 'threatened', as threatened has a clearer definition and is more widely understood. Unless imperilment is used to refer to the difference between total species richness and threatened species richness. If so, this should be made explicit in the abstract and the main text.	We have now replaced 'imperilled' with 'threatened' in the title.
Careful use of 'imperilment' vs 'threatened species' is required throughout the manuscript it would be best not to use them interchangeably.	We have made the appropriate changes throughout the manuscript
14 – It might be informative to suggest what types of environmental factors and habitat characteristics are	We have now included more specific details in the abstract.

typically more important than anthropogenic impacts for explaining spatial variation in numbers of threatened species. E.g., environmental factors, such as temperature seasonality and precipitation, and habitat characteristics, such as insularity, are typically more important ... The findings in the abstract are currently very generally and more specific details would be useful for the readers.	
15 – Personally, we prefer the use of the term ‘human’ instead of ‘anthropogenic’, as we think using the word human is more readable and the word anthropogenic often leads to more detached thinking that reduces the feeling of our collective human responsibility. Up to the authors though.	We have now replaced ‘anthropogenic’ with ‘human impact’ throughout.
16 – We would mention the focal taxa in the abstract for clarity, so instead of saying ‘these relationships vary between taxa’ you could say ‘these relationships vary between terrestrial vertebrate groups’ or similar.	We have included the reviewer’s suggestion in the abstract.
21 – The spaces before references should be removed throughout the manuscript.	Spaces have been removed
43 – It could be easier for the reader if the order of ‘predisposing environmental conditions’ and ‘anthropogenic threatening processes’ is consistent throughout the manuscript, including the order they are introduced (line 37). For us, it makes most sense to describe or refer to the predisposing environmental conditions first and the extrinsic threatening processes second.	We agree with the reviewer regarding the order in which to refer to the two processes and have ensured that we refer to them in this order throughout. We have not, however, changed the order in which they are introduced. We feel that the threatening processes need to be introduced first, before we can introduce the environmental conditions that predispose species to the effects of those threatening processes can be described.
56 – Consistent use of environmental/habitat/energy/anthropogenic variables is needed. Which variables count as environmental, which as habitat related? In the abstract (line 14) they are described as environmental, habitat and anthropogenic, whereas in the Results and Discussion (line 67) they are referred to as habitat, energy and anthropogenic. Also for Figure 2 and Figure 3.	We now consistently refer to ‘Environmental’ and ‘Human Impact’ covariates throughout.
73 – For us, Supplementary Figure 3 does not equivocally confirm the idea that island regions support a higher number of endemic, extinction-prone species than continental land masses, as insularity shows a bimodal distribution. Therefore, we would suggest dropping reference to Supplementary Figure 3 here.	Apologies, the labels for habitat diversity and insularity had been mixed up. We believe that this plot now offers better evidence to support this idea.
75 – Check use of superscript for 1st, 2nd, 3rd, etc. Currently some are superscript and some are not.	Apologies, we have now corrected this
93 – What is the word ‘significantly’ here based on? If	We have now deleted ‘significantly’

it's not based on a statistical test the term 'significantly' could be dropped.	
106 – We would suggest defining 'vagile' or using a simpler expression, as the readership of this journal is broad.	We have replaced 'vagile' with 'mobile' (Line 99)
195 – Need a reference for the source of the elevation data.	Apologies for this oversight, reference now included (Line 217).
275 – It would be useful to state the values of m and nt that were actually used to fit the final set of models.	A full model summary is now included in the supplementary materials (Table S1 and S2).
291 – Consistency is needed for the use of 'zoogeographic' vs 'biogeographic' realms. We assume that the zoogeographic realms referred to in the statistical analyses (line 291) are the biogeographic realms described previously (line 223). Also for Figure 3.	We now use the term 'zoogeographic region' throughout. This is in line with Holt et al., (2013)
294 – Which models lacked explanatory power, which regions?	A full model summary is now included in the supplementary materials (Table S1 and S2).
478 – Figure 2, duplication of information for types of variables. Either drop letters or drop colours. If letters are kept they need to be described in the Figure caption.	We have dropped the colours in favour of the letters, which we now define in the legend.
487 – Figure 3, for me, decreasing mean R2 would make more sense, as the best explained models would appear first.	We now present the regional-level models ranked by decreasing R ² in Figure 3.
487 – Figure 3, why does the model for the Japanese Biogeographic/Zoogeographic realm have low explained variance? Might be nice to suggest why this might be in the main text.	This is because the Japanese region is smaller than the other regions and had fewer models converge. This is now made clear in Supplementary Table S1.
495 – Figure 4, why were the selected variables selected. A short description of the selection criteria would be informative.	We originally presented variables with contrasting relationships with total and threatened species richness. We now present environmental and human impact covariates which are of high importance in the taxa specific global models of threatened species richness. We have added this justification for the variables selected in the methods section (Lines 335 - 339). We also present all relationships in the Supplementary Materials (Figures S10 and S11).
Supplementary Table 2 – There is no description of this ANOVA in the methods. Needs to be described in the main or Supplementary methods	We now describe these ANOVAs in lines 325 - 328.
Supplementary Figure 3 – y-axis label is misspelt, richness is spelt as 'richnes'	Apologies, this has now been corrected.

References

1. Cutler, D. R. *et al.* Random forests for classification in ecology. *Ecology* **88**, 2783–2792 (2007).

2. Nicodemus, K. K., Malley, J. D., Strobl, C. & Ziegler, A. The behaviour of random forest permutation-based variable importance measures under predictor correlation. *BMC Bioinformatics* **11**, 110 (2010).
3. Kier, G. *et al.* A global assessment of endemism and species richness across island and mainland regions. *Proc. Natl. Acad. Sci.* **106**, 9322–9327 (2009).
4. Collen, B., Ram, M., Zamin, T. & McRae, L. The Tropical Biodiversity Data Gap: Addressing Disparity in Global Monitoring. *Trop. Conserv. Sci.* **1**, 75–88 (2008).
5. Franklin, J. *Mapping Species Distributions: Spatial Inference and Prediction*. (Cambridge University Press, 2010).
6. Valavi, R., Elith, J., Lahoz-Monfort, J. J. & Guillera-Arroita, G. blockCV: an R package for generating spatially or environmentally separated folds for k-fold cross-validation of species distribution models. *Methods Ecol. Evol.* **00**, 1–8 (2018).
7. Radosavljevic, A. & Anderson, R. P. Making better Maxent models of species distributions: complexity, overfitting and evaluation. *J. Biogeogr.* **41**, 629–643 (2014).
8. Hijmans, R. J. Cross-validation of species distribution models: removing spatial sorting bias and calibration with a null model. *Ecology* **93**, 679–688 (2012).
9. Breiman, L. Random forests. *Mach. Learn.* **45**, 5–32 (2001).
10. Araujo, M. B., Pearson, R. G., Thuiller, W. & Erhard, M. Validation of species-climate impact models under climate change. *Glob. Chang. Biol.* **11**, 1504–1513 (2005).
11. Segurado, P., Araujo, M. B. & Kunin, W. E. Consequences of spatial autocorrelation for niche-based models. *J. Appl. Ecol.* **43**, 433–444 (2006).
12. Roberts, D. R. *et al.* Cross-validation strategies for data with temporal, spatial, hierarchical, or phylogenetic structure. *Ecography (Cop.)*. n/a-n/a (2016). doi:10.1111/ecog.02881
13. Sinha, P. *et al.* Assessing the spatial sensitivity of a random forest model: Application in gridded population modeling. *Comput. Environ. Urban Syst.* **75**, 132–145 (2019).
14. Holt, B. G. *et al.* An Update of Wallace’s Zoogeographic Regions of the World. *Science (80-.)*. **339**, 74 LP – 78 (2013).

Tables and Figures

Table R1: Comparison of model performance metrics. The reported coefficient of determination is the mean R^2 from across the ten random forest models. This value is derived from predictions made to a spatially segregated test data block omitted from model training. These are the values reported in the manuscript. The internal pseudo R^2 values are mean R^2 values from the random forest reporting outputs. These values are derived during the internal cross validation performed when fitting the random forest models.

Taxonomic Group	Reported Coefficient of		Internal Pseudo	
	Determination (R^2)	\pm S.D.	R^2	\pm S.D.
Amphibian	0.72	0.054	0.81	0.005
Bird	0.92	0.007	0.96	0.000
Mammal	0.94	0.012	0.97	0.001
Reptile	0.87	0.025	0.92	0.002
Total Vertebrates	0.94	0.012	0.96	0.001

Figure R1: Importance of individual variables for predicting global threatened species richness, including data deficient species richness as a covariate. Top panel (a.) indicates the importance of individual variables from the global models of vertebrate threatened species richness, whilst the bottom panels indicate the measures of individual variable importance from the global models of amphibian (b.), reptile (c.), bird (d.) and mammal (e.) threatened species richness. Variables are grouped into broader classes, which are indicated by the capital letters on the side of the variable names: Environmental (E), Human impact (H), and Other (O) covariates. Variables are ranked by their median importance in the model of vertebrate threatened species richness. The line across each box indicates the median and the box boundaries indicate the interquartile range (IQR). Whiskers identify extreme data points that are not more than 1.5 times the IQR on both sides; the dots are more extreme outliers.

Reviewers' comments:

Reviewer #1 (Remarks to the Author):

The authors have addressed most previous comments well. In some instances issues remain.

In response to an earlier comment the authors state they address the potential for circularity between insularity and threatened species richness in lines 128-129. The text reads "This is surprising, given the importance of insularity in the global models and the role of geographic range size in classifying species' extinction risk". This circumvents the issue. It fails to mention the potential circularity in analyses that test the importance of a predictor that is known to be used to define the response. I agree the predictor should be considered but the potential circularity pitfall needs to be clearly stated.

I still find that figure 1 does not show patterns for amphibians very well, if there are any. The scale is mostly orange and red colors but do not see these in the map that had blue and grey primarily. Maybe the red areas are very tiny and indeed there is little spatial variation. Figure S1 (and some others in the supplementary materials) suffer from the same problem. I suggest a different approach to define breaking points for color categories, clearly there are many categories with very few values and thus maps are rather uninformative.

For figure 4 I remain confused about the criteria used to select the two variables represented per group. Initially I thought there would be the top two variables or top H and top E, but that is not the case (why not?). For amphibians and reptiles, invasives species was more important than the chosen H variable (for reptiles the E variable was also not the most important E variable). For birds and mammals the H variable is actually one with low importance. A rationale for the choice of variables would improve this work. My suggestion would be displayed the top H and top E variable for each.

The non-linearity of species richness (and threatened richness) in mammals with human activity/land appropriation (line 166) has been shown by Polaina et al (2018. *Global Ecology and Biogeography* 27: 647-657 DOI: 10.1111/geb.12725) where mechanisms aligned to those discussed here were proposed, that previous work (which is also relevant as a previous study looking at global drivers of mammalian threatened richness) should be recognized and cited.

Finally two new suggestions.

Results are primarily discussed focusing on H vs E variables. However, figures 2 and 3 do not show variables in these groups together, which makes it harder to see the overall group patterns. For me it would be easier to interpret results if variables were sorted by importance but within groups, so first O, then E and then H and within each from highest to lowest. As presented I found it took time

to see which was the most important H variable or how H variables ranked compared to E overall (there are plenty of low importance E variables).

In line 177 the text reads “Our global assessment of the drivers of threatened species richness reveals that those environmental characteristics that predispose species assemblages to threat are far more influential than threatening human processes for determining where concentrations of imperilled species occur. This finding has important implications for conservation planning. Knowing the inherent vulnerability of a region can aid decisions regarding global conservation priorities and could form the basis for a biodiversity conservation roadmap”. I wonder if the text should mention that regional findings (which show some cases where human impacts can be important) could also be useful to identify priorities and vulnerabilities. Environmental factors are difficult to manipulate or control, whereas arguably human activities can be more easily changed by humans.

Reviewer #2 (Remarks to the Author):

The authors have done a good job of addressing all of my original concerns with additional analyses or discussion. I haven't reviewed their responses to the other reviewers, but I am happy that my part of the review has been satisfactorily dealt with.

Marcel Cardillo

Reviewer #3 (Remarks to the Author):

This manuscript presents spatial patterns and predictors of total species richness and threatened species richness for terrestrial vertebrates. Overall, we think this is an interesting study, with findings that are of broad relevance to a wide conservation audience.

We previously reviewed this paper and believe it is now much improved, particularly the communication of the findings and the sensitivity analyses, which support the validity of the results. We are also convinced by your additional justification of the your random forest analyses, and the cross-validation approach you have taken.

We now only have a few minor points for consideration. The most important of these relates to what we feel is an underplaying of the effects of human predictors on threatened species (below), particularly given the caveat the authors have now added that acknowledges that the coarse-grain nature of the analysis favours climatic over human disturbance variables.

Line edits:

60 – We feel that a qualifier might be needed when presenting the main finding of the greater importance of environmental than human predictors of threatened species richness. The wrong inference to make would be that human threats are unimportant and have negligible influence on threatened species richness, which would be easy to infer from what is written (particularly line 70). In fact, long term land cover change and invasive alien species have high variable importance (just lower than the environmental variables; Fig. 2a), which is discussed later. Moreover, the new section on the impacts of the spatial grain of the analysis (lines 117-124) makes it clear that the grain of these analyses is likely to underestimate the full effect of human disturbance relative to environmental variables. As such, the key take home is potentially misleading and could be misused/misquoted. Instead, we'd recommend something along the lines of "Global models of threatened vertebrate species richness revealed a greater influence of environmental parameters than extrinsic human threats (Figure 2a, Tables 1 and 2), although human threats still had an important role." This more nuanced message is also needed in the abstract.

177-180 – Again, this statement is too strong given you are working with 0.5 degree data, and given that the human influence variable still come through as being pretty important throughout. Please revise this statement accordingly.

180 – "Knowing the inherent vulnerability of a region can aid decisions regarding global conservation priorities and could form the basis for a biodiversity conservation roadmap. For example, areas that are inherently at greater risk from the effects of threatening human processes could be prioritised for stricter protective actions whilst mitigation - by remediation, for instance - may be more appropriate in less vulnerable areas." We're not sure this is what was actually tested. There's no assessment of which areas are more sensitive to human pressures. The areas of higher threatened species richness might be more sensitive to threatening processes, and this seems like a reasonable assumption, but you have no direct evidence for this. It could also be the case that threatened species are less sensitive to human pressure, for example, naturally rare species may have greater adaptive responses for coping with small population sizes. In brief, we think it should be clarified as a suggestion, e.g., "For example, areas that are inherently more imperilled might also be at greater risk from the effects of threatening human processes, and therefore could be prioritised for stricter protective actions whilst mitigation ..."

197 – "may occur in future" should be "may occur in the future"

210 – "Total and total threatened species richness ...". I would spell out the indices in full e.g., "Total species richness and total threatened species richness ...". Total and total makes for difficult reading

355 – Need full GitHub address

527 – The aspect ratio for sub-panels b-e in Figure 1 is incorrect, leading to distorted maps (i.e., they are stretched tall; compare the shapes of the continents in sub-panel a, which look correct, and panels b-e). We understand that space is restricted, however the current presentation leads to distorted patterns and unfair representation of areas. This is also true for the Supplementary spatial figures (S1-S3), where the global maps also look distorted (Fig. S1a, S2a, S3a).

Response to Reviewers: NCOMMS-19-15718A

Reviewer Comment	Suggested Response
Reviewer 1	
In response to an earlier comment the authors state they address the potential for circularity between insularity and threatened species richness in lines 128-129. The text reads “This is surprising, given the importance of insularity in the global models and the role of geographic range size in classifying species’ extinction risk”. This circumvents the issue. It fails to mention the potential circularity in analyses that test the importance of a predictor that is known to be used to define the response. I agree the predictor should be considered but the potential circularity pitfall needs to be clearly stated.	We have now added a statement to the methods to detail the potential circularity in our analysis (Lines 229 - 235).
I still find that figure 1 does not show patterns for amphibians very well, if there are any. The scale is mostly orange and red colors but do not see these in the map that had blue and grey primarily. Maybe the red areas are very tiny and indeed there is little spatial variation. Figure S1 (and some others in the supplementary materials) suffer from the same problem. I suggest a different approach to define breaking points for color categories, clearly there are many categories with very few values and thus maps are rather uninformative.	We have now re-plotted Figure 1 using Jenks natural breaks¹ instead of a continuous colour scale. We believe that this better emphasises the spatial distribution of residual threatened species richness that is displayed in Figure 1. We have also plotted out the data displayed in Figure 1 using a deciles approach to define the break points (Figure R1 at the end of this response). We believe that the Jenks natural breaks provides a more even spread along the species richness scale than the deciles approach. If, however, the editor feels that the deciles approach provides a better illustration of the distribution of residual threatened species richness, we are happy to use Fig. R1 instead. We have also replotted Figures S1 and S3 using a log transformed colour scale, which we believe better shows the spatial variation in threatened, total and data deficient species richness patterns.
For figure 4 I remain confused about the criteria used to select the two variables represented per group. Initially I thought there would be the top two variables or top H and top E, but that is not the case (why not?). For amphibians and reptiles, invasives species was more important than the chosen H variable (for reptiles the E variable was also not the most important E variable). For birds and mammals the H variable is actually one with low importance. A rationale for the choice of	Each variable included in Figure 4 is one of the three most important environmental and human impact variables for each taxonomic group, for which the response of total species and threatened species richness differs. We have not included variables for which the relationship with total and threatened species richness is the same, as these variables may be acting as proxies for the effects of total species richness on threatened species richness. Having already identified total species richness as a

variables would improve this work. My suggestion would be displayed the top H and top E variable for each.	major determinant of threatened species richness patterns (Figure 2) we do not believe that these relationships are of notable interest. We do, however, present all relationships in the supplementary materials (Figures S10 and S11). We have now included additional justification of the variables selected for inclusion in Figure 4 (Lines 351 - 353).
The non-linearity of species richness (and threatened richness) in mammals with human activity/land appropriation (line 166) has been shown by Polaina et al (2018. Global Ecology and Biogeography 27: 647-657 DOI: 10.1111/geb.12725) where mechanisms aligned to those discussed here were proposed, that previous work (which is also relevant as a previous study looking at global drivers of mammalian threatened richness) should be recognized and cited.	We now include reference to Polaina et al (2018)² on Line 172.
Results are primarily discussed focusing on H vs E variables. However, figures 2 and 3 do not show variables in these groups together, which makes it harder to see the overall group patterns. For me it would be easier to interpret results if variables were sorted by importance but within groups, so first O, then E and then H and within each from highest to lowest. As presented I found it took time to see which was the most important H variable or how H variables ranked compared to E overall (there are plenty of low importance E variables).	As suggested by the reviewer, we have now ordered the variables in Figures 2 and 3, and all relevant supplementary figures, first by group and then by importance in the global model.
In line 177 the text reads “Our global assessment of the drivers of threatened species richness reveals that those environmental characteristics that predispose species assemblages to threat are far more influential than threatening human processes for determining where concentrations of imperilled species occur. This finding has important implications for conservation planning. Knowing the inherent vulnerability of a region can aid decisions regarding global conservation priorities and could form the basis for a biodiversity conservation roadmap”. I wonder if the text should mention that regional findings (which show some cases where human impacts can be important) could also be useful to identify priorities and vulnerabilities.	We have now added text to emphasise the importance of the regional and taxonomic level results for conservation planning (Lines 183 - 184). However, in recognition of points raised by reviewer 3, we have refrained from adding text that would suggest that management of human activities should be focused in areas where we show human impacts to be of greatest influence. When our results suggest that environmental conditions are of the greatest influence, we believe that they imply that environmental processes are driving threatened species richness through the promotion of small range, endemic species that are more susceptible to the effects of human threatening processes. Ultimately, it is human activities that are often the drivers of species’

Environmental factors are difficult to manipulate or control, whereas arguably human activities can be more easily changed by humans.	extinctions, and as such they should be managed in all regions. We believe that our results can provide insight into how susceptible a region is to, and the level of protection an area requires from, threatening human processes.
Reviewer 3	
60 – We feel that a qualifier might be needed when presenting the main finding of the greater importance of environmental than human predictors of threatened species richness. The wrong inference to make would be that human threats are unimportant and have negligible influence on threatened species richness, which would be easy to infer from what is written (particularly line 70). In fact, long term land cover change and invasive alien species have high variable importance (just lower than the environmental variables; Fig. 2a), which is discussed later. Moreover, the new section on the impacts of the spatial grain of the analysis (lines 117-124) makes it clear that the grain of these analyses is likely to underestimate the full effect of human disturbance relative to environmental variables. As such, the key take home is potentially misleading and could be misused/misquoted. Instead, we'd recommend something along the lines of "Global models of threatened vertebrate species richness revealed a greater influence of environmental parameters than extrinsic human threats (Figure 2a, Tables 1 and 2), although human threats still had an important role." This more nuanced message is also needed in the abstract.	We have altered the text according to the suggestions made by the reviewer (Lines 61 - 62). We have also adapted the abstract to include the more nuanced message identified by the reviewer (Lines 8 - 9).
177-180 – Again, this statement is too strong given you are working with 0.5 degree data, and given that the human influence variable still come through as being pretty important throughout. Please revise this statement accordingly.	We believe that this is an important point. We have both revised the statement identified by the reviewer and added text to emphasise the importance of the human influence variables (Lines 179 - 184).
180 – "Knowing the inherent vulnerability of a region can aid decisions regarding global conservation priorities and could form the basis for a biodiversity conservation roadmap⁸. For example, areas that are inherently at greater risk from the effects of threatening human processes could be prioritised for stricter protective actions whilst mitigation - by remediation, for instance - may be more	We have altered the text following the suggestion made by the reviewer (Lines 187 – 188).

appropriate in less vulnerable areas.” We’re not sure this is what was actually tested. There’s no assessment of which areas are more sensitive to human pressures. The areas of higher threatened species richness might be more sensitive to threatening processes, and this seems like a reasonable assumption, but you have no direct evidence for this. It could also be the case that threatened species are less sensitive to human pressure, for example, naturally rare species may have greater adaptive responses for coping with small population sizes. In brief, we think it should be clarified as a suggestion, e.g., “For example, areas that are inherently more imperilled might also be at greater risk from the effects of threatening human processes, and therefore could be prioritised for stricter protective actions whilst mitigation ...”	
197 – “may occur in future” should be “may occur in the future”	We have now changed this in line with the reviewer recommendation (Line 203).
210 – “Total and total threatened species richness ...”. I would spell out the indices in full e.g., “Total species richness and total threatened species richness ...”. Total and total makes for difficult reading	We have changed this as suggested (Line 216).
355 – Need full GitHub address	We have omitted this so far to allow for a double-blind peer review. The full GitHub address has been provided to the editor.
527 – The aspect ratio for sub-panels b-e in Figure 1 is incorrect, leading to distorted maps (i.e., they are stretched tall; compare the shapes of the continents in sub-panel a, which look correct, and panels b-e). We understand that space is restricted, however the current presentation leads to distorted patterns and unfair representation of areas. This is also true for the Supplementary spatial figures (S1-S3), where the global maps also look distorted (Fig. S1a, S2a, S3a).	We have now corrected this distortion in Figures 1, S1, S2, and S3.

References

1. Jenks, G. F. Visualizing statistical distributions and generalizing process. in *Annals of the Association of American Geographers* vol. 57 179 (Blackwell, 1967).
2. Polaina, E., González-Suárez, M., Kuemmerle, T., Kehoe, L. & Revilla, E. From tropical shelters to temperate defaunation: The relationship between agricultural transition stage and the distribution of threatened mammals. *Glob. Ecol. Biogeogr.* **27**, 647–657 (2018).

Figure R1: Global variation in threatened species richness, having accounted for total species richness. The maps show the mean residuals from 10 global scale models of threatened species richness when explained by total species richness, for a) terrestrial vertebrates, and four separate taxonomic groups: b) amphibians, c) reptiles, d) birds and e) mammals. The colour scale indicates model residuals in terms of the number of threatened species, note the scale differs between panel a, and panels b – e. Grey indicates areas where there are no species classified as threatened. Model performance was moderate (Table 1 and Supplementary Table S2).